# Pico-Sat to Ground Control: Optimizing Download Link via Laser Communication

**Revital Marbel** [1] , **Boaz Ben-Moshe** [1] **and Tal Grinshpoun** [2,*]

1 Department of Computer Science, Ariel Cyber Innovation Center and Astrophysics, Geophysics & Space Science Research Center, Ariel University, Ariel 4070000, Israel; revitalm@ariel.ac.il (R.M.); benmo@g.ariel.ac.il (B.B.-M.)
2 Ariel Cyber Innovation Center, Department of Industrial Engineering & Management, Ariel University, Ariel 4070000, Israel
* Correspondence: talgr@ariel.ac.il

**Abstract:** Consider a constellation of over a hundred low Earth orbit satellites that aim to capture every point on Earth at least once a day. Clearly, there is a need to download from each satellite a large set of high-quality images on a daily basis. In this paper, we present a laser communication (lasercom) framework that stands as an alternative solution to existing radio-frequency means of satellite communication. By using lasercom, the suggested solution requires no frequency licensing and therefore allows such satellites to communicate with any optical ground station on Earth. Naturally, in order to allow laser communication from a low Earth orbit satellite to a ground station, accurate aiming and tracking are required. This paper presents a free-space optical communication system designed for a set of ground stations and nano-satellites. A related scheduling model is presented, for optimizing the communication between a ground station and a set of lasercom satellites. Finally, we report on SATLLA-2B, the first 300 g pico-satellite with basic free-space optics capabilities, that was launched on January 2022. We conjecture that the true potential of the presented network can be obtained by using a swarm of few hundreds of such lasercom pico-satellites, which can serve as a global communication infrastructure using existing telescope-based observatories as ground stations.

**Keywords:** lasercom; pico-satellite; free-space optical communication; new space



## 1. Introduction

The need for reliable global coverage IoT networks is constantly growing. A wide range of applications, including precise agriculture, search and rescue, and remote sensing, require global coverage. 4G–5G networks can support very high bandwidth, yet the coverage is mostly limited to populated regions, such as cities, roads, and industrial zones, which only constitute a minor portion of Earth's terrain (and oceans). Satellite communication is commonly used in order to achieve global coverage. In recent years, several major "new space" projects were suggested and implemented with global coverage capabilities. New-space projects, such as Starlink, Iridium-Next, GlobalStar, and OneWeb [1], are all using radio frequency (RF) licensed band as the main means of communication.

In this paper, we suggest an alternative cost-effective laser-based communication link between a low Earth orbit (LEO) nano-satellite and an optical ground station. The suggested solution allows a greater bandwidth and does not require any global licensing. Yet, a natural drawback of directional networking in general and optical transmission in particular is the obvious need to have a line of sight (LOS) between the transmitter and the receiver. Free-space optical communication (FSO) is known to be sensitive to atmospheric loss and light pollution. Another challenge related to FSO is the need to aim the transmitter and the receiver towards each other with high accuracy. Nevertheless, there is an ever-growing demand for bandwidth and an increasing need for cost-effective global networking. As a consequence, the use of laser communication (FSO) in best-effort service

may be applicable for several store-and-forward use-cases, and specifically in cases in which there is a need for high bandwidth with no real-time restrictions. Figure 1 illustrates the main two use-cases of FSO satellites from both the satellite's and the ground station's point of view.

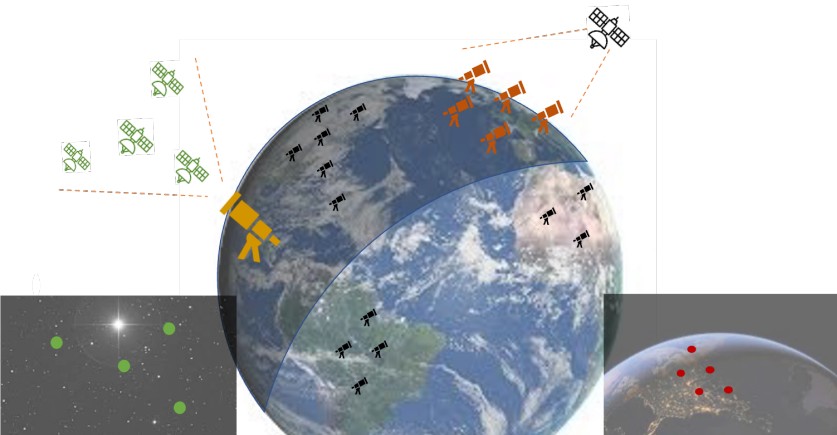

**Figure 1.** A general illustration of the satellite–terrestrial FSO communication resource allocation and scheduling model. On the (**left** side), a ground station's point of view is presented: the orange telescope should be aligned to one of the four (green) nano-satellites. On the (**right** side), a nano-satellite's point of view is shown: it should point to one of the five (red) ground stations.

This paper offers a comprehensive view of the two sides of satellite–terrestrial FSO communication. The first side is of the ground station. In this context, we describe the components that enable the required functionality of the ground station. We then explain the steps to initiate and maintain an FSO communication link with a satellite. Additionally, there are situations in which the LOS temporarily disappears, e.g., due to local weather conditions that obscure the satellite; we explain how tracking can be maintained in such situations via position prediction of the obscured satellite. We also discuss a scheduling problem that arises when the ground station needs to communicate with multiple satellites.

The second side of the communication is of the satellite. As in the case of the ground station, we also describe the satellite's components that enable its required functionality. We then explain the FSO communication steps from the satellite's viewpoint. Additionally, we provide a link-budget formulation, as well as performance evaluation. Table 1 demonstrates the expected (average) performance of various communication technologies for nano-satellites (RF vs. laser).

**Table 1.** Communication technologies related to nano-satellites.

| Nano-Satellite Communication: RF vs. Laser | | | | | | |
|---|---|---|---|---|---|---|
| Category | UHF | S-Band | X-Band | Ka | 1550 nm | 1064 nm |
| Bit rate (Mbps) | 0.01 | 0.5 | 5 | 20 | 1000 | 50 |
| Free to use | no | no | no | no | yes | yes |
| Suitable for 1 U | yes | yes | so so | so so | yes | yes |
| Tx beam (deg) | 90 | 30 | 10 | 3 | 0.1 | 0.05 |
| All weather | yes | so so | yes | so so | so so | no |
| Project Example | Norby | GOMX3 | GOMX3 | NSLSAT | CLICK * | AC 7-B |

* The CLICK demonstration nano-satellite is scheduled to be launched on 2023; all the other nano-satellites are "in orbit".

### 1.1. Related Work

FSO communication involves the use of modulated optical (laser) beams to send telecommunication information through the atmosphere. The concept of light communication is not new, and was first introduced by Alexander Graham Bell in 1880, who

demonstrated the use of an intensity-modulated optical beam to transmit telephone signals through the air to a distant receiver [2]. Over the last two decades, wide-band FSO links have been considered as a back-hauling technology for terrestrial and urban networks, mainly in cases where fiber optics and RF communication links are not applicable [3,4]. Fixed FSO links between buildings have long been established and form a separate commercial product segment in local and metropolitan area networks [5,6]. However, the mobile and long-range applications of this technology are aggravated by extreme requirements for pointing and tracking, due to the small optical beam divergences of FSO. Aside from the noted challenge of accurate pointing, long-haul FSO links have notable disadvantages related to the fading effect, dispersion, and signal obstruction due to weather and atmospheric conditions [7,8]. Motivated by the Optical Communications and Sensors Demonstration (OCSD) by NASA using 1.5 U nano-satellite demonstrations [9], which presented a 50–100 Mbps laser link with sub $10^{-6}$ bit error rate (without forward error correction), this paper presents a "best-effort" dynamic network framework. Thus, we assume clear-sky conditions. For comprehensive studies regarding "all-weather" FSO communication, see [2,5,7,8]. A dynamic FSO link (e.g., between a drone and an FSO ground station) suggests a major tracking challenge. Moreover, generalizing FSO links to an FSO network raises several problems regarding optimal topology and routing. To this end, NASA is launching an advanced laser communication experiment, termed Laser Communications Relay Demonstration (LCRD). The experiment will use NASA's ground stations and satellites with a payload designed to test such communication in terms of quality and atmospheric effect. The primary end-goal motivation of the project is to enable capturing high-definition data such as 4K video from space [10]. Kong et al. [11] described a ground station distributor for communication networks that involve satellites and unmanned aerial vehicles (UAVs). They suggested a hybrid FSO/RF communication model, in which a ground station receives satellite data using FSO communication and distributes them using RF communication to the end-users (UAVs). Vu et al. [12] proposed to use high-altitude platforms as relays in the communication between satellites and vehicles (either aerial or ground). Their suggested dynamic network is based on quantum key distribution (QKD) and FSO communication. More specifically, their solution uses high-altitude platforms as mobile relay stations that deliver the data from the satellite to the vehicles (end-users). The communication between the components in this network uses FSO communication as an encrypted channel and RF communication as a public channel. The secure link is based on QKD encryption.

Recently, the use of MEMS gimbals was presented in the concept of full-duplex FSO communication [13,14], allowing a significant miniaturization of FSO robotic (dynamic) sensors.

The space laser–communication research field is not limited to the Earth–space link. New space developments can also support satellite networks which are based on laser communication. NASA has already created the CubeSat Laser Infrared CrosslinK (CLICK) [15] mission to demonstrate a full-duplex optical communication crosslink between two 3 U small spacecrafts in LEO. The goal of this mission is to create a satellite constellation or swarm that carries one or more moderate- to high-resolution imaging or scientific sensors and communicates at high speed. The concept of a nano-satellite with emitting lights that should be visible by the naked eye was first implemented on the FITSat-1 nano-satellite [16]. Figure 2 presents the first LED flashing of FITSat-1 (2012). Finally, NASA's Terabyte InfraRed Delivery (TBIRD) program is designed to allow 200 Gbps laser communications system for LEO direct-to-Earth missions; recent preflight tests [17] show promising results regarding the ability of optical ground stations to track and establish laser links with FSO nano-satellites.

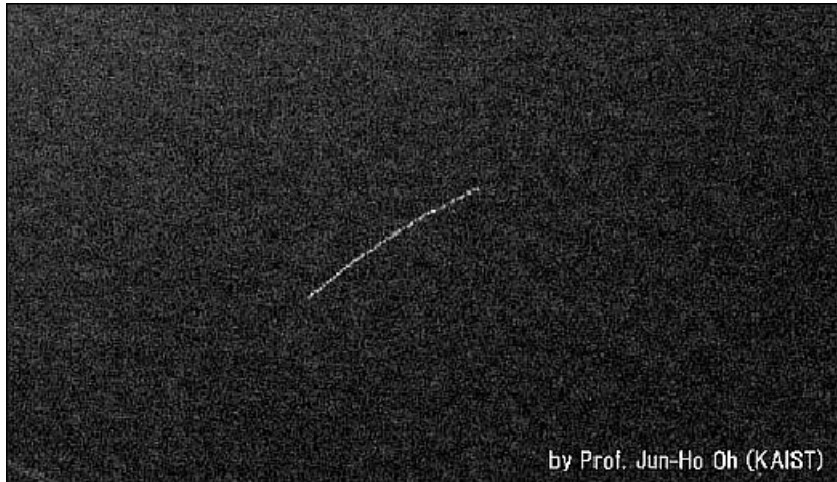

**Figure 2.** The first LED flashing of FITSat-1 as reported by Kazuhisa Mishima of the Kurashiki Science Center (Japan) and by Jun-Ho Oh of KAIST (Korea Advanced Institute of Science and Technology) (image credit: FIT).

This work describes a framework that combines FSO and RF Earth–celestial communication. It relies on new studies on space FSO that have already demonstrated the feasibility of creating an optical link in space. Kashif et al. [18] presented a link-budget analysis of a hybrid FSO-RF transmission system for communications links of up to 12 km in clear sky conditions. Alipour et al. [19] proposed an FSO system simulated at 10 Gbps bit rate and 2 km path link with different modulation types. An analysis for FSO link 1550 nm waves in 50 km was performed by Carrasco-Casado and Mata-Calvo [20]. They claim that most commercial off-the-shelf (COTS) components for fiber communications can also be used for FSO communication, especially at the optical C band, in which there are fewer absorption peaks. This is because the standard communication wavelengths (1550 nm) do not coincide with spectrum lines of strong absorption.

### 1.2. Motivation

In recent years, the number of LEO satellites is growing rapidly. Almost all the satellites are using RF communication in order to download data to Earth. Therefore, each such satellite is required to have a dedicated frequency coordination, allocated via the International Telecommunication Union (ITU). The demand for RF frequencies grows with every new satellite, which causes a bottleneck on the RF spectrum. Moreover, each country has its own RF channel regulation, which limits the location of ground stations mainly to the country of the satellite's operator. Alternatively, FSO (lasercom) requires no frequency licensing and therefore allows such satellites to communicate with any optical ground station on Earth. Laser optics use very high frequencies and therefore enable significantly faster data rates using very small antennas; such antennas commonly have extremely narrow transmission patterns, thus allowing a high-gain transmission in a particular direction. It should be stated that FSO communication is significantly more sensitive to atmospheric conditions than RF communication. Yet, we conjecture that the superior data rate of the FSO solution and the ability to use many (globally spread) ground stations can be beneficial in a wide range of remote sensing applications (e.g., the Planet [21] use-case).

### 1.3. Our Contributions

This work presents a new lasercom model for large-scale swarms of nano- and pico-satellites. To the best of our knowledge, this is the first work that suggests such an FSO network for pico-satellites. The model consists of detailed descriptions of the two types of communicating parties in the network—the ground station and the satellite. The model relates to both the physical structure of the two types of parties and to their interaction when communicating with each other. Finally, this paper includes a set of field experiments with

an operational pico-satellite (named SATLLA-2B) which demonstrates the full potential of the presented new FSO satellite network.

### 1.4. Paper Structure

This paper is organized as follows: Section 2 describes the design of the two types of communicating parties—the ground station and the satellite, including their required operations to obtain FSO communication. Section 3 presents the optimization tracking problem of scheduling a ground station to the related satellites in its site. Section 4 presents results from different perspectives—link-budget analyses, lab experiments and field experiments from our pico-satellite (SATLLA-2B). Finally, Section 5 concludes the FSO network model and discusses deployment and possible future work.

## 2. Physical Components and Methods

In this section, we describe the physical components of the two types of parties in the described network—the ground station and the satellite. We then proceed to describe the required operations by the two sides to obtain reliable FSO communication.

### 2.1. Components of the Ground Station

An FSO-based ground station requires the ability to establish a communication link with a satellite. The FSO link should be acquired in a timely manner and maintained reliably throughout a given time window. Additionally, to increase the overall capacity of the suggested FSO network, there is a need for a significant number of ground stations. Therefore, we address the need to use cost-effective COTS products that are widely available and integrated.

To answer the above requirements, we suggest herein a ground station model that is composed of the following components (see Figure 3):

- **Telescope.** Designed for tracking the satellite.
- **Laser beam with a high-speed modulator.** The laser notifies and marks the ground station's location so that the satellite can adjust its orientation directly to it. After adjusting the satellite's orientation, the optical link can be created.
- **Camera.** A wide field of view (FOV) camera for accurate tracking. The FOV of the telescope's lens is very narrow, making it hard to create the initial link and perform fast tracking accurately. However, a wide-angle camera stationed and calibrated on the telescope can detect the satellite's position and enable the telescope's tracking.
- **Laser communication transmitter and receiver.** Enables FSO communication between the ground station and the satellite.
- **Image processing.** An image processing component is required for detecting the satellite in the images of the camera and the telescope. This component enables fast-tracking and location prediction even in cases in which the satellite is hidden (e.g., clouds).

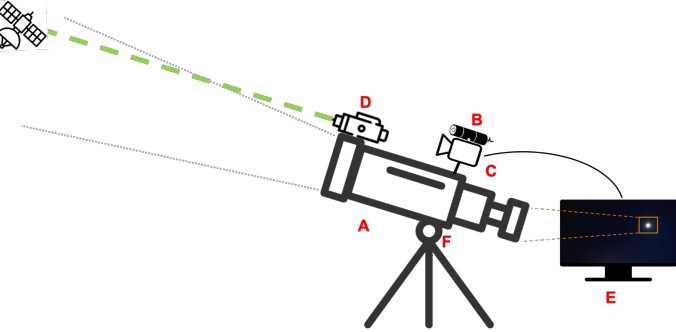

**Figure 3.** The ground station components: (A) telescope; (B) laser beam (laser beacon); (C) wide FOV camera; (D) FSO transmitter; (E) image processing component; and (F) gimbal mount with two degrees of freedom system (pitch and yaw) and aiming control.

Robotic telescopes are primarily designed for tracking celestial bodies, such as stars or planets. Hence, they are less suitable for tracking fast targets such as LEO satellites. Any robotic telescope should undergo a calibration process once (and remain fixed in place). The calibration process involves focusing the telescope lens on the target and calibrating the telescope angles to a global angle system (i.e., lat/lon). Telescopes which are designed for satellite tracking should also be able to perform relatively smooth tracking in at least 2–3 degrees per second. Yet, unlike photography telescopes that require "sub-pixel stability", telescopes for FSO applications do not need to "stand still" on the target and may be even a single milliradian off the center of the satellite, as long as it is within the field of view of the telescope, which is commonly larger than half a degree (or 8.7 milliradian). The following subsection describes the telescope's tracking algorithm in FSO-based ground stations.

### 2.2. Components of the Satellite

A nano-satellite with FSO capability requires the following three logical units: (i) standard nano-satellite platform; (ii) attitude determination and control system (ADCS); and (iii) FSO modem.

This framework is designed to support a sizeable nano-satellite constellation composed of dynamic FSO links between multiple satellites and ground stations. In order to optimize the performance and the cost-effectiveness of such FSO constellation, each unit should be designed for mass production. In particular, such a satellite should be as small as possible since the cost of its launching is determined by the satellite's weight and volume.

Each nano-satellite is equipped with standard RF communication, commonly ultrahigh frequency (UHF) with limited bandwidth. The RF modem transmits to the ground beacons containing information about the satellite's state, such as location, battery status, speed, angle, and short information regarding the data required to be downloaded. The FSO link functions for wide-band communication, particularly for transmitting extensive data, such as images and videos.

The satellite in this model, see Figure 4, has three main parts:

- **Standard satellite unit.** This unit contains (i) a lithium battery and a solar panel that provides and preserves the satellite's energy; (ii) a micro-controller board that controls the different components of the satellite. This controller is in charge of commands to start UHF or laser communication, open the camera, and track the ground station. This board is also programmed to schedule the satellite's full-power and power-save modes according to the communication demand and distance from the ground station.

- **FSO communication unit.** This unit contains two communication components: (i) UHF radio; (ii) FSO laser beam and LED light that signals to the ground station the location and orientation of the satellite.

- **Tracking unit.** This unit contains all the components that allow the satellite attitude control. It includes reaction wheels that perform the satellite's aiming towards a target, two sensors that provide the data for the tracking, a camera, and an inertial measurement unit (IMU). Accurate high-frequency tracking is accomplished via an algorithm that fuses the data from the camera's image and the self-orientation provided by the IMU.

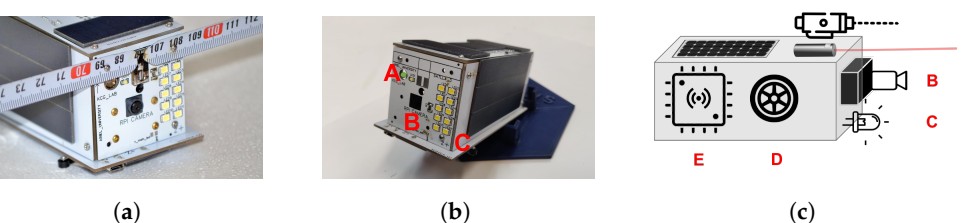

|  |  |  |
| :---: | :---: | :---: |
| (**a**) | (**b**) | (**c**) |

**Figure 4.** Images and illustration of the SATLLA2.0 nano-satellite engineering model. The left side (**a**) shows the complete engineering model of SATLLA2.0, including a UHF antenna. The image in the middle (**b**) displays the FSO components: (A) laser diode with about 1 degree beam divergence (green 532 nm); (B) Sony's IMX219 8 MP camera; and (C) an array of 10 LEDs emitting an equivalent to 20 flashes of smartphones. The satellite's emitting light should be visible from Earth using a small 8″ telescope on a clear night. The right side (**c**) illustrates the main components of SATLLA2.0: (A) FSO communication unit and a calibrated laser light; (B) camera with an image processing unit; (C) LED—for ground tracking; (D) stabilization control (reaction wheels) unit; and (E) main controller with UHF radio transmitter-receiver.

### *2.3. FSO Communication—The Ground Station's Side*

The telescope's tracking process consists of three main parts:

- Coarse tuning: adjusting the telescope angle to the area where the satellite should pass. There, it waits for the satellite to continue tracking it.
- Accurate tracking: creating a continuous communication link between the telescope and the satellite using image processing for precise attitude control.
- Continuous link: using motion prediction to create accurate, high-quality tracking.

Figure 5a illustrates the communication link between the ground station and the satellite; an actual image of the International Space Station (ISS) captured from our ground station is shown in Figure 5b.

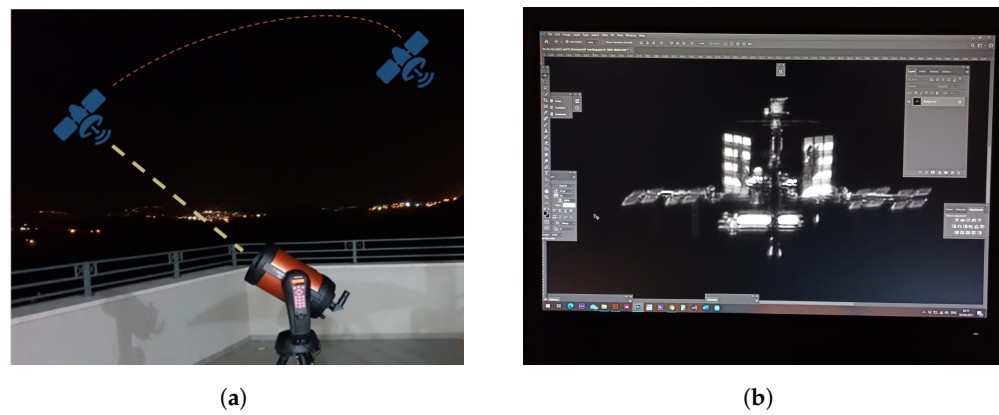

|  |  |
| :---: | :---: |
| (**a**) | (**b**) |

**Figure 5.** (**a**) Abstraction of a ground station–satellite communication link. (**b**) The ISS as captured from our ground station; shot taken by Michael Tzukran [22] on 3 June 2021.

### 2.3.1. Coarse Tuning

Recall that the ground station includes a robotic telescope and a tracking system. Robotic telescopes have a 2-DoF (degree of freedom) aiming mechanism—defined by two angles: azimuth (*Az*) and elevation (*Elv*). Assuming the telescope is calibrated, the first angle, *Az*, represents the horizontal polar angle from the north and the second angle, *Elv*, represents the elevation angle. The orbit of each satellite is given by TLE (two-line element) parameters. The North American Aerospace Defense Command (NORAD) [23] detects, monitors, and updates the TLE of each satellite (and even space debris). Given the ground station's position, the time, and a TLE (of a satellite), a robotic telescope can aim itself and follow the satellite's TLE. The coarse tuning phase begins a few seconds before the

optical link is created. At this phase, the ground station calculates where the satellite is expected to rise and aims to reach that location. In order to aim a telescope to a satellite, the relative vector between the telescope and the satellite needs to be computed. The telescope's Cartesian coordinates representation is denoted as $\{x_t, y_t, z_t\}$, and the satellite's expected Cartesian coordinates are represented as $\{x_s, y_s, z_s\}$. The equations for computing the $Az, Elv$ angles are presented next.

Let $\{x, y, z\}$

$$\Delta x = x_s - x_t \tag{1}$$

$$\Delta y = y_s - y_t \tag{2}$$

$$\Delta z = z_s - z_t \tag{3}$$

$$Elv = \tan^{-1}\left[\frac{\Delta y}{\Delta x}\right] \tag{4}$$

$$Az = \tan^{-1}\left[\frac{\Delta z}{\sqrt{\Delta x^2 + \Delta y^2}}\right] \tag{5}$$

Computing the TLE of a satellite and converting GPS-like global angular coordinates (lat, lon, alt) to Cartesian coordinates $(x, y, z)$ are two technical topics which are out of this paper's scope; see [24] for a technical guide for satellite tracking by telescopes using TLE information. The above method only aims the telescope to the expected area of the satellite's orbit. Obtaining an accurate tracking of such a small object requires an accurate TLE and an accurate pre-phase calibration of the telescope. In this work, we suggest a hybrid model that will replace the need for an accurate TLE and a well-calibrated telescope system.

2.3.2. Accurate Tracking

The main challenge of accurate satellite tracking is that a minor deviation in the telescope angle is sufficient to lose line of sight with the satellite. This is due to the telescope's narrow viewing angle, which is very narrow and designed for slow tracking. A wide FOV camera, whose function is to capture the satellite when it is out of the telescope's line of sight, enables reorientation.

To properly track the satellite, the camera should be stably mounted on the telescope and accurately calibrated to the telescope lens, i.e., the center pixel in the image of the wide FOV camera should be the same as the center of the telescope lens. This way, when the satellite appears in the center of the camera, it also appears in the telescope lens.

The tracking process is carried out automatically via the main processing unit that is connected to both the camera and the telescope. This processing unit has an image processing program that tracks the satellite movement in the image and reorients the telescope accordingly. This model relies on the unique design of the satellite with light-emitting diode (LED) that can be seen from Earth. The LED will turn on when passing a ground station in order to be seen from Earth. When the satellite appears in the frame, it resembles a twinkling star. The accurate tracking phase comprises three repetitive steps:

- Detect the satellite in the image.
- Compute deviation from the image's center.
- Reorient the telescope so that the satellite appears in the lens.

This process, if performed at a rapid rate, can ensure that the telescope is linked with the satellite throughout its pass.

**Satellite detection:** The satellite detection in the image of the wide FOV camera image is performed in two steps: (i) edge detection in the image and (ii) satellite classification.

The edge detection process can be performed using the Canny edge detection algorithm [25], which is a five-stage algorithm for finding edges in an image. The algorithm steps are as follows: apply a Gaussian filter, find the intensity gradients, apply a gradient magnitude threshold, apply a double threshold, and remove all weak and unconnected edges. This algorithm provides smooth edges by using noise reduction and accurate edges

by applying a double threshold on the image gradient. This is a common method for accurate edge detection and is implemented in almost every image processing library. In our study, we apply the Canny function implementation from the Python OpenCV library [26].

The satellite classification step determines which object is being detected. A common night-sky image may consist of many objects that can appear similar to the satellite (e.g., stars, light pollution). The distinction method relies on the fact that the satellite continuously changes its position. Using image processing on a video instead of on an image can detect the satellite's movement by comparing the images in the video and observing the differences.

Figure 6 demonstrates the satellite detection process by simulating the tracking of the ISS. The ISS passes through the night sky in a visible route (the sunlight is reflected on the ISS) once in two days on average. Knowing its predicted position (that can be learned from [27]), we were able to video its passing using our wide FOV camera. After doing so, we used the image processing algorithm to detect the station in the image.

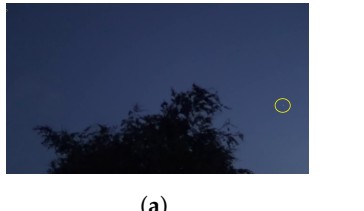 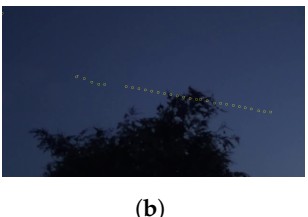 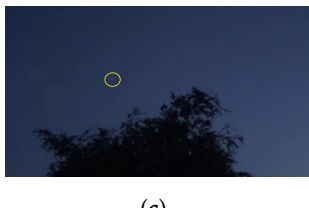

(**a**) (**b**) (**c**)

**Figure 6.** Satellite detection demonstration via video processing: (**a**) tracking the ISS at its rise; (**b**) tracking the ISS via video processing using the Kalman filter algorithm (note that the video was shot in low resolution, causing tracking breaks in several places); and (**c**) the final phase of the tracking (credit to RichardB1983 for the raw video).

**Deviation Computation:** To compute the new angles for the telescope aiming, we calculate the pixel deviation of the satellite from the image center (which is calibrated with the telescope lens) and convert these pixels to angles. Since the camera is calibrated to the telescope and perpendicular to the ground, the x-axis can be converted to *Az* angle and the y-axis to *Elv* angles.

**Telescope reorientation:** Assuming that the telescope is calibrated and stable, then by obtaining the computed angles (*Az*, *Elv*) as input, the telescope can perform the reorientation.

### 2.3.3. Prediction

The prediction phase of the communication occurs when a direct LOS is not available. Direct LOS can be broken, for example, due to local weather conditions that obscure the satellite for the entire duration of its passing or parts of it. Using the image from the ground station's wide FOV camera, we can predict the expected break duration. Image processing on these images can detect possible obstacles in the satellite trajectory. Using the data on these obstacles, the system can predict location and time of the LOS breakage. When small breaks are expected, the system is able to continue the accurate tracking process using motion prediction.

The motion prediction can be obtained by fusion of two parameters over time—(i) the satellite position in the image and (ii) the satellite's predicted trajectory computed using the previous frames' measurements. Kalman filter, named after Rudolph E. Kalman [28], is a method used for stochastic estimation from noisy sensor measurements. It describes a recursive solution to the discrete-data linear filtering problem. This filter uses a set of mathematical equations that implements a trajectory predictor corrector that minimizes the estimated error covariance. Kalman filter has been the subject of many studies and applications in the area of navigation and tracking.

In this study, we implement a simple version of the Kalman filter method for predicting the satellite location throughout the tracking. The Kalman filter method has two steps: prediction and update. In the prediction step, the system calculates the predicted location

of the satellite in time $t$, given the last measurements in $t - \Delta T$. The update step calculates the position using two parameters: the estimated position of the satellite in time $t$ based on the new measurements (the satellite detection in the image) and the predicted position calculated in the first step. The updated position is then calculated using a covariance matrix that measures the estimated uncertainty of the prediction. The covariance matrix determines the weight of the predicted location in the final estimation. Next, we present the required equations for each step $k$.

**The prediction equations:** Let $X_{k-1} = [x_{k-1}, y_{k-1}, vx_{k-1}, vy_{k-1}]$ be the predicted position and velocity vector. Let $P_{k-1}$ be the covariance matrix calculated in step $k - 1$. Let $A_{k-1}$ be the matrix that represents the motion equation of the satellite in the image by which the prediction is calculated. In this model, we also assume a constant velocity for each $\Delta t$:

$$A = \begin{bmatrix} 1 & 0 & \Delta t & 0 \\ 0 & 1 & 0 & \Delta t \\ 0 & 0 & 1 & 0 \\ 0 & 0 & 0 & 1 \end{bmatrix}$$

The satellite prediction equations:

$$X_k^p = AX_{k-1} + Q \tag{6}$$

The predicted state is calculated from the previous state update and the estimated satellite motion that is given by the matrix $A$. $Q$ are used as tuning parameters that can be adjusted to achieve the desired performance.

$P$ represents the estimated accuracy of the state measurement:

$$P_k^p = AP_{k-1}A^T \tag{7}$$

The matrix $P$ in the initial state is the zero matrix, since we know the satellite's initial location with high accuracy.

**The update equations:** Let $Z = [x_m, y_m, vx_m, vy_m]$ be the measured location and velocity vector of the satellite in the image. The Kalman gain matrix is denoted as $K$. Let $R_k$ be the matrix representing the noise in the measurements $R_1 = I_{4 \times 4}$.

$$K = P_k A^T (AP_{k-1}A^T + R_k)^{-1} \tag{8}$$

$$P_{k+1} = P_k - KAP_k \tag{9}$$

The new position $X_{k+1}$ is then calculated:

$$X_{k+1} = AX_k + K(Z - AX_k) \tag{10}$$

This final equation calculates the satellite's location according to the prediction and the measurement values. This equation gives more weight to the predicted value when the Kalman gain is small, which means that in such cases, the system relies less on the measurements.

**A note on TLE-based tracking:** Recall that the orbit of a satellite is represented using TLE (two-line element); whenever an accurate TLE is available, it can serve as a single source of tracking. Yet, in many cases, nano-satellites are hard to detect and identify and therefore it might take several weeks to obtain a reliable TLE. Moreover, the expected accuracy of nano- and pico-satellites is relatively low, as their orbit tends to drift faster than larger satellites. In such cases, the proposed trajectory Kalman filter can be used.

*2.4. FSO Communication—The Satellite's Side*

In order to establish an FSO link, both communicating sides need to be accurately aligned towards each other. The motion of the satellite requires both the ground station

and the satellite to perform accurate tracking. The handshake process between the satellite (*S*) and the ground station (*GS*) consists of the following stages:

1. *S* performs a **coarse alignment** to *GS* and turns on the LED.
2. *GS* **detects and tracks** *S* and turns on its laser beacon.
3. *S* detects the laser (of *GS*) and **accurately aims** itself towards *GS*.
4. *S* turns on its FSO laser and starts the **data transmission**.
5. *GS* signals to *S* an **FSO feedback** with its laser beacon, using a very low data rate.

Commonly, satellites have an onboard GPS. Moreover, their orbits are determined and monitored by the Space Surveillance Network [29] and the TLEs are publicly available. Both the GPS and the TLE allow global positioning and velocity. The satellite orients itself towards Earth, searches for the telescope (the visual beacon), and aims its laser to the visual beacon (the center of the camera's FOV).

We now elaborate on each stage of the FSO link establishment process:

- **Coarse alignment.** The satellite aims itself roughly at the region of the ground station. For performing an Earth-pointing rough alignment, see [30].
- **Detection and tracking.** Figure 7 demonstrates a laser beacon ground experiment. The detection process (in this case) is rather straightforward. Yet, when the beacon is relatively weak, we use both the blinking rate and the laser's frequency (color) to increase the signal-to-noise ratio (SNR) of the laser beacon.
- **Accurate aiming.** Figure 8 shows the attitude control determination of the satellite, which is composed of three reaction wheels, an IMU, and a beacon-detection sensor (based on a wide-angle camera).
- **Data transmission.** After the satellite aligns itself with respect to the laser beacon of the ground station, it starts transmitting the data using a simple on–off keying (OOK) protocol; see [31,32].
- **FSO feedback.** The ground station may signal back to the satellite (using the laser beacon) low bandwidth information regarding the quality of the FSO link. In particular, the following types of information are passed: packet drop rate, suggested optimal modulation, and time to expected non-line-of-sight (NLOS).

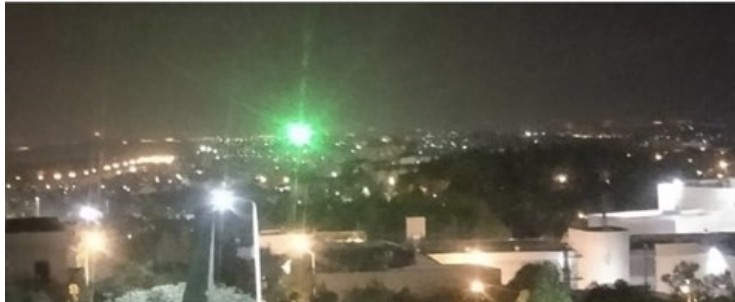

**Figure 7.** Terrestrial laser-beacon detection and tracking. A 0.002 Watt green laser as captured from ∼9 km by a basic Raspberry Pi camera in VGA mode.

Most satellites in orbit have an attitude determination and control system (ADCS), which allows the satellite to orient itself in order to aim its antenna or camera towards the Earth. It should be noted that while FSO is a means of communication, the required orientation accuracy is higher than a directional RF antenna and closer to aiming a camera—commonly better than 1 milliradian. We propose the use of micro reaction wheels to aim the FSO pico-satellite to the ground station (see Figure 8a). The main challenge of adjusting the stabilization system to FSO usage is that the FSO transceiver should point to the ground station with high accuracy and follow it while orbiting. In order to achieve such accurate aiming, this model uses the laser of the ground station as orientation feedback, instead of the self-orientation sensor (such as star-tracker). By doing so, the satellite is able to track the ground station in a close loop with the help of the mounted camera and the

IMU. The satellite angular accuracy requirement is defined by its laser divergence. Thus, assuming a 1 milliradian laser beam, a sub 0.5 milliradian angular accuracy is needed. Using a 2 megapixel camera (1080p) with 30 deg field of view, the pixel angle is about 0.25 milliradian, which is sufficient for continuous tracking, even without super-resolution; see [33] for an arcminute (about 0.3 miliradian) attitude stability on a nano-satellite.

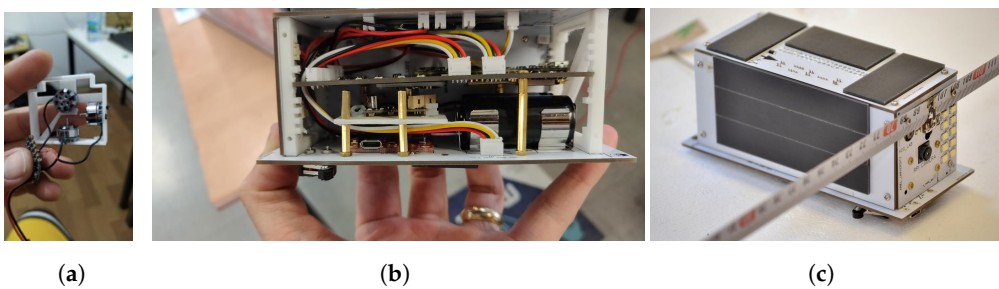

(**a**)  (**b**)  (**c**)

**Figure 8.** The FSO pico-satellite engineering model. (**a**) The attitude control unit, composed of three tiny reaction wheels, one for each axis, and a tiny flight controller with IMU for controlling each motor. (**b**) SATLLA-2.0 pico-satellite under construction. (**c**) The final model of SATLLA-2B pico-satellite, with its LEDs, camera, laser, UHF antenna, and solar panels. This model was recently successfully launched into LEO.

### 3. Satellite Scheduling by a Ground Station

In this section, we explore the algorithmic solutions for a scheduling problem that arises when a ground station needs to successively communicate with many satellites.

A ground station is expected to successively communicate with many satellites. Each satellite has its own orbit; therefore, from the viewpoint of the ground station the satellites appear (and disappear) in different times and from different angles. Moreover, it is expected that for a given ground station at a given time there might be several visible satellites, as illustrated in Figure 1. This, in turn, yields that the ground station should choose with which satellites to communicate. In many real-world situations, such scheduling choices are of high importance. For instance, a ground station may serve as a "middleman" that downloads valuable information from the different satellites and then sells this information to various buyers. The value of the downloaded data may vary considerably depending on the satellite with which the ground station chooses to communicate. Consequently, an efficient scheduling policy may lead to high revenues. This financial model can be naturally generalized to any type of utility, not necessarily monetary, that the owner of the ground station may gain from the obtained data. Planet's use-case [21] represents such a large-scale optimization and scheduling problem.

The described problem of the ground station is known in the scheduling literature as the $1|r_j|\Sigma w_j U_j$ problem. Following the $\alpha|\beta|\gamma$ notation of scheduling problems [34], 1 in the $\alpha$ field means that there is a single ground station (as opposed to a more involved many-to-many model). Each satellite (job) has a time window in which the ground station can communicate with it. In the scheduling terminology, such a time window is expressed by a *release time* and a *deadline* for each job. Having release times for jobs is expressed by the $r_j$ constraint in the $\beta$ field, whereas the existence of a deadline is implied by the objective function $\Sigma w_j U_j$ in the $\gamma$ field. $\Sigma U_j$ indicates a *unit penalty* for each job that does not meet the deadline, whereas the multiplication in $w_j$ enables job-dependent weights (utilities). The objective functions in scheduling literature are usually minimization functions. As a consequence, instead of maximizing the utility gained from successful communications with satellites, we minimize the unsuccessful (or missed) communications.

More than four decades ago, Lenstra et al. [35] proved that the $1|r_j|\Sigma w_j U_j$ problem is strongly NP-hard, even in its unweighted form ($1|r_j|\Sigma U_j$). Nevertheless, some exact algorithms were developed for this intractable problem. Péridy et al. [36] proposed a branch-and-bound algorithm that utilizes a Lagrangian lower bound. Their algorithm

managed to solve most 100-job instances in reasonable time. Slightly better results were obtained by the branch-and-check algorithm of Sadykov [37].

Despite the improvement in exact solution methods, large-scale problems require heuristic methods. Sevaux and Dauzère-Pérès [38] propose genetic algorithms for solving $1|r_j|\Sigma w_j U_j$. An advantage of genetic algorithms is that they can be easily adapted to changes in the model. For instance, Sevaux and Sörensen [39] also devised a genetic algorithm for a similar problem in which the release times of jobs are not precisely known in advance. The robustness to such variations in the problem along with the ability to solve large instances suggest that genetic or other bio-inspired algorithms are appropriate for handling the satellite scheduling problem at hand.

## 4. Results

In this section, we present both simulated-in-field results. First, we present FSO link-budget analyses between a pico-satellite and an optical ground station. Then, we elaborate on the required improvements for obtaining a 1 Gbps FSO Link. Next, we focus on both lab and field experiments that validate the applicability of our proposed model. Finally, we present experimental results related to the launched pico-satellite SATLLA-2B.

### 4.1. Link Budget for FSO Nano-Satellite

We present herein two simple link-budget formulations for FSO communication between an LEO nano-satellite and an optical ground station (as described in Section 2.1). The first formulation is based on "photon-counting" link budget (following the modeling of the "NODE" project [40]). The second formulation is based on standard RF link budget. We also provide an analysis of the overhead that stems from dealing with errors. In the following subsection, we add a generalization and improvement of the presented link-budget analysis in order to allow up to 1 Gbps laser communication link between a pico-satellite and an FSO ground station.

The following parameters were used: Beam divergence of 1 mrad and a range of 1000 km (between the satellite and the ground station) implying a 1 km$^2$ "spot". A 14″ telescope has an area of 0.1 m$^2$, leading to a geometric ratio of $Geo_{rt} = 10^{-7}$. The modeled transmitter has a bit rate of 10 Mbps and power of $Tx_{power} = 100$ mW. Assuming a clear night sky, the expected loss is around 3 dB (about 50%, denoted by $Atm_{loss}$). The telescope losses are also bounded by 50% (denoted by $Rx_{loss}$).

### 4.1.1. Photon-Counting Laser Link-Budget Analysis

- A standard 1 mW laser induces about $C_1 = 3.2 \times 10^{15}$ photons per second.
- Existing sensitive detectors, i.e., avalanche photodiode (APD), have a ratio threshold of 100–300 photons per bit [40]. In such ratio, a bit error rate of $[10^{-3}, 10^{-4}]$ is expected (without forward error correction).
- Assuming the transmitter has a bit rate of 10 Mbps and power of $Tx_{power} = 100$ mW, the transmitter emits $C_1 \times 10^2 \times 10^{-7} = 3.2 \times 10^{10}$ photons per bit.
- Based on the geometric ratio of $Geo_{rt} = 10^{-7}$, the telescope is expected to receive 3200 photons per bit in optimal conditions.
- The atmospheric losses ($Atm_{loss}$) and the telescope losses ($Rx_{loss}$) are bounded by 50%, allowing about 800 photons per bit; this should be sufficient for a robust and high-bandwidth FSO link between a nano-satellite and a ground station.

In order to reduce the "light-pollution" (or the "photon-noise") in the ground stations, 1500 nm lasers can be used. In such frequencies, there is a low light pollution and there are available (COTS) sensitive existing photon detectors. We refer the readers to the CLICK [15], NODE [40], and DeMi [41] projects related to satellite laser communication. These projects simulate and demonstrate the main two challenges of lasercom: pointing (tracking) and optical link optimization.

### 4.1.2. Photon-Counting LED Link-Budget Analysis

- A single Lumen equals $4.09 \times 10^{15}$ photons per second.
- An LED, as used in smartphones' flash and in the discussed pico-satellite-SATLLA-2B, induces about 60 Lumen (lm), which is $L_1 = 60 \times 4.09 \times 10^{15} = 2.4 \times 10^{17}$ photons per second.
- SATLLA-2B has an array of 10 LEDs, such a combined array induces about 600 lm which is $L_{10} = 600 \times 4.09 \times 10^{15} = 2.4 \times 10^{18}$ photons per second.
- The angle of the LED FoV (field of view) is about 60 degrees and the satellite distance from Earth is about 500 km ($500, 000$ m). Yet, in order to perform an FSO communication, a distance of 1000 km is assumed, which leads to an area of $10^{12}$ m$^2$.
- The telescope width is assumed to be 14″, which has an area of about 0.1 m$^2$.
- Therefore, the geometric ratio between telescope area and the emitted area on ground is about $10^{-13}$.
- Based on the geometric ratio, the telescope is expected to receive $2.4 \times 10^{18} \times 10^{-13} = 2.4 \times 10^5$ photons in optimal conditions.
- The atmospheric loss ($Atm_{loss}$) and the telescope loss ($Rx_{loss}$) are assumed to be about 3 dB (50%) each (or about 25% combined), allowing about $2.4 \times 10^5 \times 0.25 = 6 \times 10^4$ photons per seconds received in the the telescope detector.
- Using a sensitive optical detector (with expected ratio of 100 photons for bit, and 20% header overhead), a bit rate of 500 bits per second is expected.
- Finally, a sensitive camera mounted on the telescope should be able to detect the pico-satellite and estimate the actual FSO path loss and link budget.

### 4.1.3. RF Link-Budget Analysis

We now present the link-budget analysis using RF modeling:

**Link Margin:** The link margin is a calculation that shows the likelihood of creating wireless communication under certain conditions. In our case, we show the possibility of creating a link between relatively simple communication components given optimal conditions, i.e., a clear night sky without clouds, rain, or haze. Our calculation mainly considers the geometric position, the elevation angle, and the receiver's sensitivity and transmission power. The link margin is computed for two wavelengths: 850 mm and 1550 mm. The link-budget equation is

$$LinkMargin = Tx_{power} - R_s - G_{loss} - Tur_{loss} - Atm_{loss} \tag{11}$$

where $Tx_{power}$ is the transmitter power, $R_s$ is the receiver sensitivity, $G_{loss}$ is the geometric attenuation, $Tur_{loss}$ is the turbulence attenuation, and $Atm_{loss}$ is the atmospheric attenuation (assuming a clear night with no clouds, rain, or haze).

**Parameters:** In this model, the $Tx_{power}$ is 100 mW, i.e., 20 dBm, and the receiver sensitivity $R_s$ is $-65$ dBm. The geometric attenuation can be calculated using this formula:

$$G_{loss} = 10 log_{10} \left( \frac{\frac{\pi}{4} (d\varphi)^2}{R_a} \right) \tag{12}$$

where $R_a$ is the receiver area (0.1 m$^2$, representing a 35 cm or 14″ telescope). Assuming that there is a satellite located 1000 km from the telescope, then there is an expected 8 dB margin for FSO communication. Table 2 presents the link budget, margin parameters, and total link loss.

**Table 2.** Link-budget parameters.

| Parameter | Units | Value | Remarks |
| --- | --- | --- | --- |
| Wavelength | nm | 850 | can be 532, 1550 |
| Link Distance | km | 1000 | |
| Tx Divergence | mrad | 1 | |
| Elevation Angle | degree | 45 | |
| Transmit Power ($Tx_{power}$) | dBm | 20 | |
| Receiver Sensitivity ($R_s$) | dBm | −65 | APD |
| Receiver Area ($R_a$) | m$^2$ | 0.1 | 14″ telescope |
| System Loss ($Rx_{loss}$) | dB | 3 | |
| Geometrical Loss ($G_{loss}$) | dB | 70 | $10^{-7}$ |
| Atmospheric Loss ($Atm_{loss}$) | dB | <3 | clear night sky |
| Turbulence Loss ($Tur_{loss}$) | dB | <1 | clear night sky |
| Total Budget | dB | 85 | |
| Total Loss | dB | <77 | |
| **Link Margin** | **dB** | **>8** | |

Based on both analyses (photon counting and RF path loss), we conclude that on clear nights a 14″ robotic telescope with a sensitive APD receiver should be sufficient to receive a bit stream of 10 Mbps with a bit error rate lower than $10^{-4}$, allowing FSO packets of size 120 bytes to have a packet error rate of 10%; such error rate can be handled using modern FEC (forward error correction) methods. Taking into account the additional FEC and packet headers, this results in a reliable bit stream of 6–8 Mbps.

### 4.1.4. FSO Overhead Analysis

As in any satellite communication, FSO communication also requires a forward error correction (FEC) code. Moreover, a proper FSO protocol should have several modulation configurations that can be rapidly adapted to allow real-time modulation optimization, since any cloud in the sky may change the link budget significantly. Assuming standard on–off keying (OOK) protocol, the following parameters should be defined for each modulation: packet size (e.g., 100, 1000, 10,000 bytes), FEC overhead (e.g., 10%, 20%, 40%), and basic bit rate (e.g., 1, 10, 100, 1000 Mbps). Assuming a bit error rate (BER) of $10^{-4}$, the use of very small packets (e.g., 100-byte packets) will be required in order to obtain a packet error rate (PER) lower than 10%. With the packet header overhead and the required FEC overhead (say 20%), the actual "good-put" speed is expected to be 70–75% of the theoretical FSO link, which is still significantly faster than an RF-based solution.

### 4.2. Reaching 1 Gbps FSO Link from a Pico-Satellite

In this subsection, we present the needed improvements to the FSO link in order to reach the 1 Gbps data-rate goal (from a pico-satellite to an FSO ground station). Reaching a 1 Mbps data rate from a nano-satellite using existing RF technology is very demanding and may require dedicated RF licensed frequencies. To further improve the FSO link, and in particular to allow a higher bit rate, the link budget should be improved. This can be achieved using the following improvements:

1.  Using high-efficiency InGaAs-biased detectors with a noise-equivalent power (NEP) as low as $2 \times 10^{-15}$ (*Watt* per $Hz^{0.5}$) at 1550 nm. Such detectors have a typical rise time of ~100 picoseconds allowing up to 5 GHz detection (suitable for 1 Gbps data

rate). Additional 5× speedup bit rate may be obtained through improved quantum efficiency (i.e., less photons are required per bit).

2.  A larger 30″-lens telescope with a high-end narrow band-pass filter to minimize the noise. Using such a telescope may increase the SNR significantly, allowing an additional speedup of 4×–5× times.

3.  Finally, using a narrower (0.5 milliradian) and stronger (say, 1 Watt) laser beam, an additional 5× speedup may be achieved.

Combined, an approximate ∼100× time speedup can be achieved. Naturally, this goal requires a massive engineering effort, which is not expected to be completed anytime soon. Yet, all the improvements mentioned above are based on existing components. Moreover, NASA was able to demonstrate a ∼5 Gbps laser link [42]. Finally, the improved link budget can also be used to allow low-speed FSO communication in non-perfect conditions, such as light pollution and long-range links.

*4.3. Performance Evaluation of SATLLA-2B*

We performed several experiments in order to validate the methods. Naturally, performing any experiment regarding nano-satellites requires significant resources. Thus, we employed the following methodology:

*   Performing an independent experiment on each component, e.g., ground station, satellite, swarm of satellites.

*   Ground station photon efficiency: A lab experiment was performed in order to validate the expected loss on the telescope. A known source of light was aimed towards the telescope (the laser beam was entirely within the telescope's field of view). On the telescope, a fiber optic was attached to the receiver adaptor. The received power at the fiber was measured to be at most 3 dB lower than the original laser power. We conclude that the overall power loss of the telescope can be assumed to be smaller than 3 dB (no more than 50%; in most cases, an efficiency of above 70% was achieved).

*   Ground station tracking: The tracking was performed using the ISS as a moving target (see Figure 5b). It is important to mention that two independent ISS tracking experiments were performed in suboptimal atmosphere conditions (central Israel in June and July 2021). In both experiments, a calibrated robotic telescope was able to track the ISS orbit using only time-based reference (without optical tracking). This allowed us to conclude that the expected aiming errors due to atmospheric turbulence are minor and have very limited effect on the performance of FSO from a ground station (the overall turbulence errors were lower than 0.01 degrees, whereas the telescope's field of view is 0.5–1.0 degrees). See [43] for detailed research regarding methods for adapting FSO links to atmospheric turbulence. Based on the ability to accurately track the satellite and assuming a clear night sky, the turbulence loss ($Tur_{loss}$) is expected to be bounded by 1 dB.

*   Performing ground experiments: We conducted a set of FSO ground experiments. These experiments were performed without a telescope in order to simulate a greater distance (from the satellite). Both the $Atm_{loss}$ and $Tur_{loss}$ were equal to or greater than expected in the satellite FSO scenario (which assumes a clear night sky, as opposed to the "noisy" ground conditions).

*   Using existing FSO system analysis: As presented in the related works, satellite FSO has already been demonstrated several times. Therefore, we mainly focus on generalizing existing solutions, so as to make them applicable for swarms of pico-satellites and a corresponding large set of optical ground stations.

**Short-range FSO:** In this experiment [44], a low-power (1 mW) red laser was positioned about 1.8 km from the receiver, causing a "spot" of 10 m² (square meters). The receiver had an active area of 1 mm² (square millimeters), which formed a $10^{-7}$ geometric ratio. The overall atmospheric path loss and Rx loss were also comparable with the expected losses from a satellite. During the experiment, a 42 kHz signal was transmitted and received using

a COTS APD detector, with an additional margin. Using a 500-times-larger optics (500 mm telescope), the range can be proportionally extended to 900 km. Moreover, by using a 1000-times stronger Tx power, the bit rate can be extended by a square-root factor, leading to an expected bit rate of above 1 Mbps. Note that this is a very conservative estimation, as the experiment setup can be improved dramatically with a better detector and upgraded optics and lasers. A longer-range experiment was repeated in a 9 km range scenario, as shown in Figure 7, resulting in similar performance evaluation under slightly harder atmospheric conditions (the 9 km range had an atmospheric loss of about 6 dB—twice the expected loss). In the short-range experiments, we modeled the expected satellite to ground station FSO parameters, in particular, $G_{loss}$, $Rx_{loss}$, $Atm_{loss}$, and $Tur_{loss}$. Moreover, despite actual losses of the $Atm_{loss}$ and $Tur_{loss}$ parameters being greater than expected, an FSO link was successfully established with the expected performance. We then tested the ability of a ground station to detect an array of LEDs as located on SATLLA-2. The engineering model of SATLLA-2B was used to see if it can be detected from 1–10 km in good conditions (see Figure 9. Thus, we conjecture that using a 500 mm telescope with an FOV of 0.8 degrees should be able to detect SATLLA-2B from a 1000 km (on clear nights) as it has a $100\times$ larger lens diameter and 1% the FoV. We conclude that the link budget as presented above implies a valid FSO communication estimation. Thus, an FSO link from a pico-satellite to an optical ground station can be obtained in optimal (clear night sky) conditions.

**Long-range tracking:** In these experiments we tested the capabilities of the ground station to track an LEO satellite. Naturally, we used the ISS as the tracking target. Figure 6 demonstrates the ability of a wide FOV camera to detect and track the ISS. Figure 5 demonstrates the actual ability of a calibrated robotic telescope to capture an LEO satellite with high accuracy.

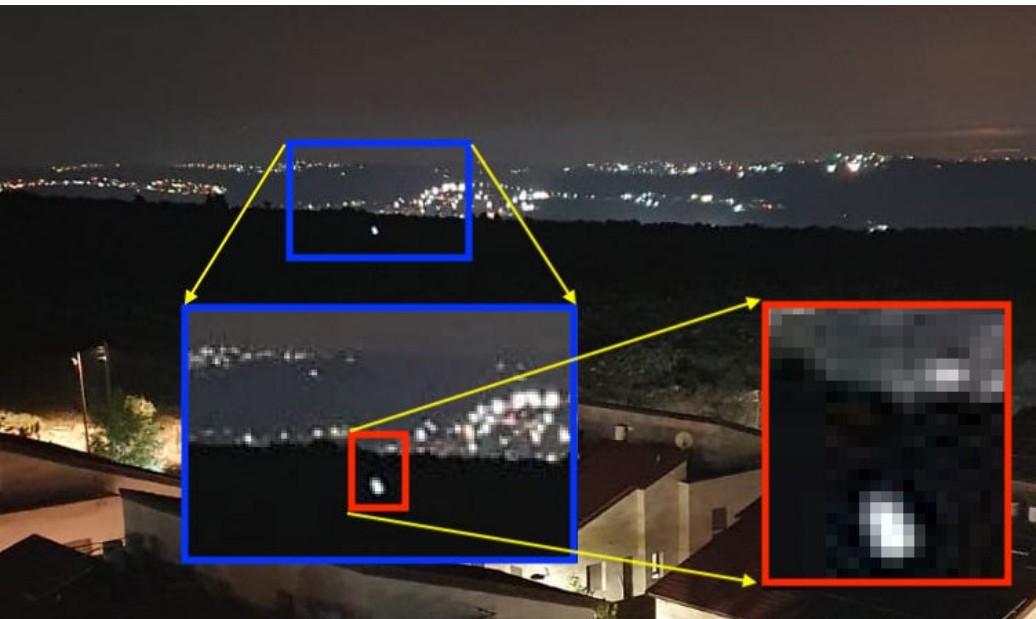

**Figure 9.** The LED array of SATLLA-2 (Eng-model) as seen from 1.1 km. The image was taken with a standard 70-degree smartphone camera—shooting at 30 fps in 1080p video format. It should be noted that in low-light-pollution conditions (on clear nights), SATLLA2 can be detected from more than 10 km using a standard Raspberry Pi V3 camera (FoV = 80, Sony IMX519).

*4.4. SATLLA-2B's Flight Result*

In this subsection, we present the preliminary results of SATLLA-2B from its low Earth orbit. The satellite was launched on January 2022 and was operational for 5 months. SATLLA-2B's angular speed is about 6–7 degrees per seconds, allowing a balanced thermal state of the satellite.

Figure 10a presents the angular velocity of SATLLA-2B without any ADCS—as it rotates naturally from its initial launch (inertia). Such a low angular velocity implies that the required force needed to overcome the pico-satellite's momentum is minimal, and in fact can be constructed using a set of miniature reaction wheels (see Figure 8a). Figure 10b presents the temperature of the satellite both on the outer and in the inner parts of the satellite. Figure 10 presents some sensory data as sensed by the satellite.

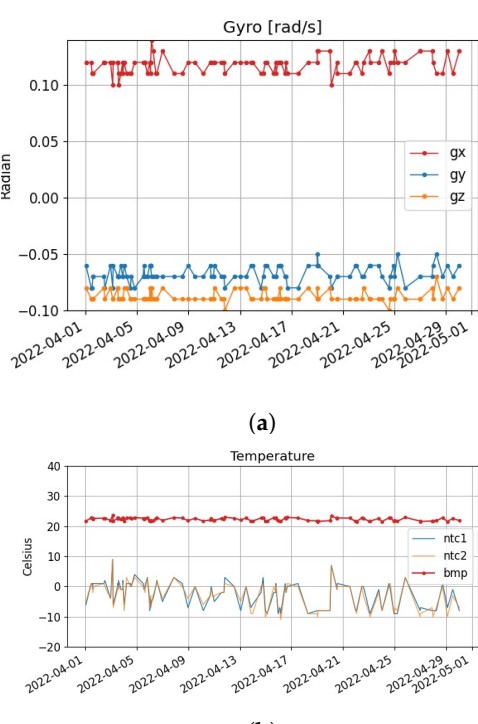

(**a**)

(**b**)

**Figure 10.** (**a**) The angular speed of SATLLA-2B in each axis, as captured by the onboard MEMS-gyro. (**b**) The temperature (in Celsius) of the satellite as sensed in the batteries ($rtc_1$, $rtc_2$) and in the satellite's IMU (*bmp*).

Several preliminary experiments are performed on this pico-satellite in order to test the FSO demonstration payload. It should be noted that the general concept of global RF communication (via LoRa protocol) was already demonstrated successfully, using the wonderful initiative of "TinyGS" [45]. Figure 11 displays a real RF beacon packet that was transmitted by SATLLA-2B and was captured by 15 amateur (RF) ground stations in a distance exceeding 1500 km.

In the first few weeks after its launch, the TLE of SATLLA-2B was inaccurate. Nonetheless, about two months after launch, the TLE became stable and well defined. On average, about 30–60 messages transmitted from SATLLA-2B are received daily by various ground stations and about 4–10 messages transmitted by ground stations are received by SATLLA-2B. Due to the narrow bandwidth (62.5 kHz) and the high speed of the satellite (∼28,000 km/h), a large portion of the messages from the satellite are received with errors, as a result of significant Doppler shift errors. It should be noted that using 125 or 250 kHz bandwidth should eliminate most of these CRC errors. The RF uplink performance of the satellite is somewhat better than was expected according to simulation results, as the actual uplink to the satellite is ∼2–3 dB better than expected; contrary to that, the download link is ∼1–2 dB lower than simulated. We argue that both gaps between the real and the simulated results are due to the RF noise level (in space it is very low, while on the ground it is relatively high). The overall cooperation with the TinyGS community is outstanding; they even allowed us to incorporate several relay experiments.

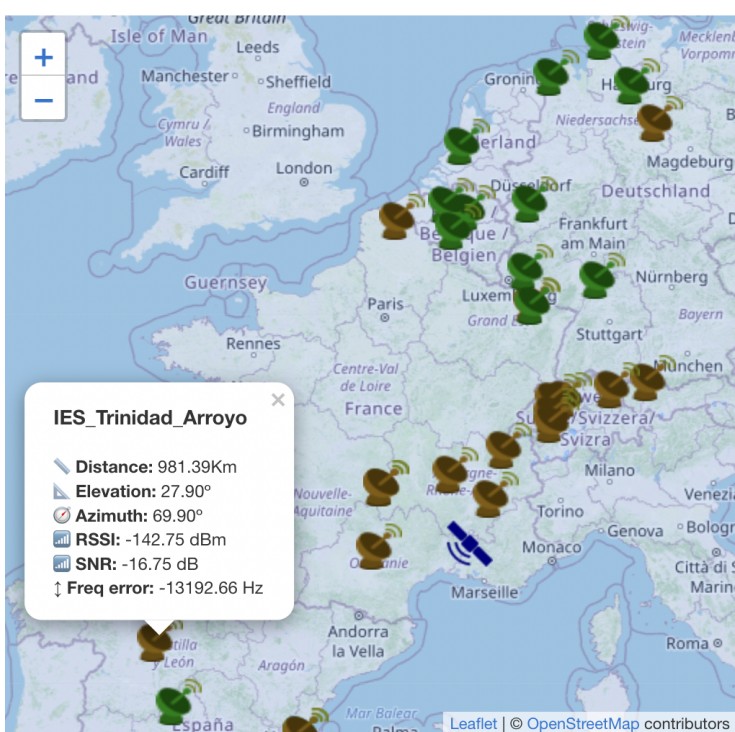

**Figure 11.** Early experiments with the recently launched SATLLA-2B pico-satellite. Using the TinyGS project, ground stations can receive and share RF data transmitted by LoRa based satellites. In this image, about 30 ground stations are receiving an RF beacon transmitted by SATLLA-2B (in a distance range of 550–2100 km). The ground stations marked in green received the beacon properly, while the brown one received CRC error; SATLLA-2B is marked in blue.

Negative Results of SATLLA-2B

While the performance of SATLLA-2B in terms of RF communication and energy optimization can be considered as a success (it truly exceeded our expectations), several key components are not functioning properly, in particular the onboard GPS of SATLLA-2B is not working (most probably due to a faulty SD card); thus, we are relying on the satellite's TLE which is not updated on a recent basis. Moreover, due to reliability concerns, SATLLA-2B was not equipped with an ADCS, making the detection of the LED array a tricky task. We were able to operate the LED flashing mechanism successfully (based on the current reading of the satellite), yet until this time we were not able to detect SATLLA-2B's LEDs. Therefore, we are currently constructing an improved ground station (18″ telescope with the rapid tracking of LEO satellite capabilities). Moreover, we are planning to open the ability to turn on the satellite LEDs to relevant observatories to help us detect SATLLA-2B (equivalent to the concept of TinyGS but with FSO links).

**5. Discussion and Future Work**

This paper introduced a new concept of FSO network for nano-satellites from the points of view of both the ground station and the satellite. The actual design concepts of an FSO pico-satellite (named SATLLA-2B, 2P, 300 g, see Figure 4) and a corresponding optical ground station were presented in Section 2. On 13 January 2022, SATLLA-2B was successfully launched into LEO orbit and is currently operational.

The scheduling problem that we discussed in Section 3 is from the viewpoint of the ground station. The same model can be applied from the point of view of the satellite that searches for ground stations to communicate with; thus, the problems are dual. Each such side of the problem is one-to-many. However, the full-scale problem is many-to-many, as it consists of many satellites that need to communicate with many ground stations. Such a problem is obviously harder than the one-to-many versions. Nevertheless, we believe that bio-inspired algorithms can serve as a good solution for such complex problems, due

to their robustness to model variations and their ability to handle large-scale problems. For example, an ant colony algorithm has been recently proposed for a related problem of network formation for FSO satellite communication [46].

The massive growth in the number of commercial and research satellites is leading to a frequency demand on the already limited RF spectrum. In order to support large sets of satellite constellations while reducing the usage of the RF spectrum, we propose the following satellite communication framework:

1.  Most satellites will not communicate directly with Earth, but will instead use ISM channels (e.g., LoRa 2.4 GHz) in order to perform inter-satellite low-speed communication to FSO satellites.
2.  The FSO satellites will aggregate the packages received by the RF ISM channel and will perform a rapid FSO download when in view of an FSO ground station. From the ground station, the packets will be routed to each satellite operator via the internet. By using the 2.4 GHz spectrum, the inter-satellite communication will impose very limited RF noise on terrestrial communication, since 2.4 GHz communication has a significant loss in the atmosphere.
3.  The upload channel may stay "as is" (based on RF channels), as the upload communication is commonly (significantly) narrower than the download communication.

The network may use application layer encryption in order to maintain the privacy of data. Naturally, such a framework will require changes in the International Telecommunication Union (ITU) licensing procedures (mainly allowing inter-satellite RF communication). Yet, it should be beneficial for both satellite and terrestrial operators. It should be noted that the presented framework cannot support real-time communication, as it may only work on clear-sky nighttime conditions. Yet, the ability to use hundreds of existing telescopes located globally, without any need for regulations, may lead to a new means of satellite communication, which is most suitable for remote sensing applications such as the Planet [21] use-case. For future work, we would like to generalize the FSO network to be able to handle real-time cases such as full duplex communication between two remote points on Earth. The concept can be implemented by allowing two close pico-satellites to intercommunicate using RF link, based on existing long-range (ISM) WiFi at 2.4, 5, or 60 GHz. Existing commercial drones allow a 10–20 km full-HD video link (in LOS conditions)—such a communication link can be used to connect two close satellites. Maintaining this range (formation flight) is yet another open question for the future [47]. The FSO uplink from the ground station to the satellite can be applied using a high-power laser on the ground station's side with low beam divergence and a 3–4 cm lens on the satellite's side; see [32].

We conjecture that by using few such lasercom pico-satellites, the true potential of the presented network can be explored via remote sensing observations by existing telescope-based observatories. The expected laser and RF communication radius of such pico-satellites is 1500 km (see Figure 11), allowing a theoretical coverage area of about $4 \times 10^6$ km$^2$. As the Earth's surface is about $510 \times 10^6$ km$^2$, a theoretical "perfect-spread" swarm of 128 pico-LEO satellites may cover the Earth. Naturally, the realistic number required for a global LEO swarm is larger—as the actual service radius of each satellite might be reduced to 1000 km and real-life layout of an LEO constellation requires a significant overlapping coverage. See [48,49] for optimization considerations of large LEO constellations. Based on simulation and the use-case of Starlink (as of Q1 2022 they have over 1700 operational LEO satellites), a swarm of about 1000 such pico-satellites can support partial coverage of 20–40% of the globe; full global coverage can be achieved with 3000–5000 pico-satellites. All this can be achieved with the price tag of just a single "real" satellite (weighing around 1 ton). Naturally, there are many other open questions regarding the orbit optimization of the swarm and the regulation of strong lasers performing long-range FSO. Finally, the increasing popularity of new-space projects, combined with the reduction of the launch cost, leads to a growing number of nano-satellites. The demand for RF frequencies grows with every new satellite, which causes a bottleneck on the RF spectrum. Therefore, offloading satellite communication to FSO seems to be a natural

solution [50]. The use of MEMS-mirrors to accurately aim an FSO laser in a sub-millisecond manner can enable a single nano-satellite to perform a time-divided communication with several FSO ground stations [51]. Naturally, implementing such a demanding system in a nano-satellite encapsulates several major research and engineering challenges. Yet, with the continuing improvement in the onboard computation capabilities and the miniaturization of key components such as ADCS and star-trackers [52], these technological barriers may be birched. We plan to follow this challenge with an annual new FSO pico-satellite. Our current model (named SATLLA-2I [53]) is scheduled to be launched in Q1 2023, and is planned to have both a star-tracker and an ADCS, allowing it to perform initial FSO communication demonstrations.

**Author Contributions:** Conceptualization, B.B.-M.; Data curation, R.M. and B.B.-M.; Formal analysis, B.B.-M. and T.G.; Funding acquisition, B.B.-M.; Investigation, R.M.; Methodology, R.M., B.B.-M. and T.G.; Project administration, B.B.-M. and T.G.; Resources, B.B.-M.; Software, R.M.; Supervision, B.B.-M. and T.G.; Validation, B.B.-M. and T.G.; Visualization, R.M.; Writing—original draft, R.M., B.B.-M. and T.G.; Writing—review & editing, B.B.-M. and T.G. All authors have read and agreed to the published version of the manuscript.

**Funding:** This research was partially supported by the Ariel Cyber Innovation Center in conjunction with the Israel National Cyber Directorate in the Prime Minister's Office.

**Data Availability Statement:** Information regarding the SATTLA projects can be found in the following google drive repository: https://photos.google.com/share/AF1QipNuD-Qzsa8XET7KWuuSVa6I 79b01jApVwpQga_sCzwHYqTrf3iwMuiCbA8I5Vkulg?key=M2UtUnZGZWlibzRXV1VBOFJ6TnBSSzd EWHYtNTV3. Also, this website contains additional information: https://www.ariel-asc.com/blog.

**Acknowledgments:** The authors would like to thank Liat Rapaport for helping with both hardware setup and field experiments. The authors would like to thank Michael Tzukran for introducing us to the real world of satellite imaging and sharing with us the amazing image of the ISS, as captured in our ground station. The authors would like to thank Rony Ronen and Michael Britvin for helpful discussions regarding the delicate engineering of designing and constructing an FSO pico-satellite. The authors would like to thank the TinyGS project, which allows us to capture our satellite data using hundreds of ground stations located around the globe.

**Conflicts of Interest:** The authors declare no conflict of interest.

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
