# Peer review of "Pico-Sat to Ground Control: Optimizing Download Link via Laser Communication"

_remotesensing, doi:10.3390/rs14153514_

Round 1
Reviewer 1 Report
I accept all of the modifications and corrections performed by the authors.
Author Response
The paper was revised according to the editor's and the other reviewers' comments.
The changes are marked in blue.
Reviewer 2 Report
This manuscript is a review article for the space laser communication used between Pico-Sat and ground station. Although detail descriptions of different components in Pico-Sat and ground station are discussion; there is no laser communication performance comparison, such as data rates, BER, wavelength, distance, etc. As a review paper, a table comparing different systems should be included.
Besides, authors have shown the photos of their SATLLA2.0; however, there is no architectures and experimental results of their work.
Author Response
The paper was revised according to the comments raised.
Below is the response to each comment (the changes are marked in blue).
Comment #1:
"Although detailed descriptions of different components in Pico-Sat and ground station are discussion; there is no laser communication performance comparison, such as data rates, BER, wavelength, distance, etc. "
Ans:
Additional reference ([9] lines 68-71) was added to the paper, it presents real laser demonstration results (bandwidth, BER, etc’). An additional “negative result” subsection (4.4.1, page 19) was added to the paper - explaining the current status of the experiments with the satellite. See answer to comment #3, for all the related information available (regarding SATLLA-2B).
Comment #2:
"As a review paper, a table comparing different systems should be included."
Ans:
A table comparing RF vs Laser communication solutions was added to the introduction (page 2)"
Comment #3:
"Besides, authors have shown the photos of their SATLLA2.0; however, there is no architectures and experimental results of their work."
Ans:
We are planning to publish most of the technical work regarding SATLLA-2. as an open-source project.
Currently, we have the following online technical manuals:
- SATLLA Blog: https://www.ariel-asc.com/blog
- How to construct: https://www.ifixit.com/Guide/SATLLA+2+DIY+KIT+Assembly/147004
- The current TLE (of SATLLA-2B): https://in-the-sky.org/spacecraft.php?id=51014
Once the github (open-source)project will be ready we will share a link to it from our blog - it will include all the above kinks - and all the needed instructions for compiling the code and contributing to it.
Regarding the experimental results:
The revised version includes additional information regarding the flight results (subsection 4.4) including “negative results” (4.4.1) and “future work” regarding our next model (SATLLA-2I) which is scheduled to be launched in Q1 2023 (page 20).
Reviewer 3 Report
In the very first sentence of the introduction, it is not clear what IoT has to do with this article (unless a swarm of satellites itself should already qualify as an example of the IoT).
Similarly, the first sentence of the abstract is a teaser ("at least once at day"). Nowhere in the article is there any calculation that refers to this problem statement. In general, the abstract has too much of an introductory character and does not adequately inform the potential reader of the hard facts that are presented (or not presented) in the paper.
The typography of formulas and of quantities needs to be fixed thoughout the paper.
$n$ in formula (4) has the dimension of a length, not an angle.
line 220: relies, not relays
line 127: out probably should read our?
The paper covers a wide range of methods and algorithms, some of which are elaborated much beyond the required detail (tracking algorithm 1, for example; by the way, in the third line, it should probably read satellite instead of telescope), and some of which are referred to in rather generalised terms. In my opinion, filters that have been the subject of textbooks for decades no longer warrant a citation of the original publication (Kalman 1960), citing a good and fairly recent review or textbook would be more helpful to the reader.
My concerns concentrates to the subject of signal-to-noise ratio and uncertainties (formerly known as errors) or accuracy. One essence of a lengthy calculation seems to be that the telescope should receive enough photons (signal) from the satellite per unit time to be detected. It remains unclear how many not-wanted photons (noise) the telescope receives during the same unit time and how the distinction between signal and noise shall be achieved. This may not be much of a problem, maybe this point caught my attention because I am not an expert in light detection, having more experience with electrical noise.
More concerning is the accuracy of the pointing which is treated rather superficially in the last paragraph of page 11. Assuming that the vertical scale of Fig. 10 (a) should read rad/s as hinted in the title, obviously the IMU's data are so inaccurate that it does not play a role in fine positioning. Let us assume that the acquisition of the target by the camera works perfectly, it then boils down to the question how fast and how accurately the reaction wheel system can respond and what amount of noise one has to expect here. The final sentence on page 11 seems almost unbelievable and is not covered by a citation (the website planet.com is not helpful at all, please provide a deep link). Again, I am not an expert in AOCS, rather in propulsion, but a quick search on AOCS precision indicates that the claim that Planet's nanosats should have a 10 microradian pointing accuracy (which would be 100 times the required 1 milliradian accuracy) needs to be substantiated for the non-expert reader. I would have loved to research this in further detail before submitting the review, but the extremely tight review schedule means that I have to pass the burden of proof back to the authors here.
Author Response
The authors would like to thank the reviewers for their helpful comments and insights. The paper was revised according to the comments raised. Below is a response to each comment (the changes are marked in blue). Please note that while the comments address the original version the answers are addressing the revised version.
Comment #1: In the very first sentence of the introduction, it is not clear what IoT has to do with this article (unless a swarm of satellites itself should already qualify as an example of the IoT). Similarly, the first sentence of the abstract is a teaser ("at least once at day"). Nowhere in the article is there any calculation that refers to this problem statement.
Ans: An additional explanation regarding the required number of LEO picosatellites for global coverage was added to the “Discussion and Future Work” section (see lines 706-712). Additional two references regarding large LEO constellations ([48],[49]) were added.
Comment #2: In general, the abstract has too much of an introductory character and does not adequately inform the potential reader of the hard facts that are presented (or not presented) in the paper.
Ans: We have tried to follow the journal's “how to write an abstract” guidelines: “We strongly encourage authors to use the following style of structured abstracts, but without headings: (1) Background: Place the question addressed in a broad context and highlight the purpose of the study; (2) Methods: briefly describe the main methods or treatments applied; (3) Results: summarize the article’s main findings; (4) Conclusions: indicate the main conclusions or interpretations.”
In case the conjecture in the last part of the abstract is too general - we can remove it altogether, yet, it seems to us as a reasonable conclusion of the presented work.
Comment #3: The typography of formulas and of quantities needs to be fixed thoughout the paper. $n$ in formula (4) has the dimension of a length, not an angle.
Ans: The formulation of the paper was rewritten to be more coherent. In particular, the azimuth elevation calculation was rewritten.
Comment #4: line 220: relies, not relays. line 127: out probably should read our? -
Ans: both typoes were fixed.
Comment #5: The paper covers a wide range of methods and algorithms, some of which are elaborated much beyond the required detail (tracking algorithm 1, for example; by the way, in the third line, it should probably read satellite instead of telescope), and some of which are referred to in rather generalised terms. In my opinion, filters that have been the subject of textbooks for decades no longer warrant a citation of the original publication (Kalman 1960), citing a good and fairly recent review or textbook would be more helpful to the reader.
Ans: Subsection 2.3 (The ground station - side) was rewritten, Algorithm1 was replaced with a simpler method. Several additional references were cited - in order to cover the technical aspects of tracking a satellite by a robotic telescope. The references to a Kalman filter were updated to include applications and a review paper (see pages 7,8).
Comment #6: My concerns concentrates to the subject of signal-to-noise ratio and uncertainties (formerly known as errors) or accuracy. One essence of a lengthy calculation seems to be that the telescope should receive enough photons (signal) from the satellite per unit time to be detected. It remains unclear how many not-wanted photons (noise) the telescope receives during the same unit time and how the distinction between signal and noise shall be achieved. This may not be much of a problem, maybe this point caught my attention because I am not an expert in light detection, having more experience with electrical noise.
Ans: Additional explanation on “photon-counting” was added (see lines 470-475), additional two papers were added to the CLICK and MeMi projects, these papers cover in detail both the pointing and the link budget challenges of laser communication.
Comment #7: More concerning is the accuracy of the pointing which is treated rather superficially in the last paragraph of page 11. Assuming that the vertical scale of Fig. 10 (a) should read rad/s as hinted in the title, obviously the IMU's data are so inaccurate that it does not play a role in fine positioning. Let us assume that the acquisition of the target by the camera works perfectly, it then boils down to the question how fast and how accurately the reaction wheel system can respond and what amount of noise one has to expect here. The final sentence on page 11 seems almost unbelievable and is not covered by a citation (the website planet.com is not helpful at all, please provide a deep link). Again, I am not an expert in AOCS, rather in propulsion, but a quick search on AOCS precision indicates that the claim that Planet's nanosats should have a 10 microradian pointing accuracy (which would be 100 times the required 1 milliradian accuracy) needs to be substantiated for the non-expert reader. I would have loved to research this in further detail before submitting the review, but the extremely tight review schedule means that I have to pass the burden of proof back to the authors here.
Ans: The following text was added to the paper: “Figure 10a presents the angular velocity of SATLLA-2B without any ADCS - as it rotates naturally from its initial launch (inertia). Such a low angular velocity implies that the required force needed to overcome the pico satellite's momentum is minimal, and in fact can be constructed using a set of miniature reaction wheels (see Figure 8a)”. (see lines: 617-620). The comment regarding the accuracy of Planetlabs nano satellites was removed from the paper (due to low relevance) and instead a reference [33] to a research paper regarding the design of a Arc-minute ADCS was added to the paper (line 396).
Round 2
Reviewer 2 Report
Authors have answered all my questions. Acceptance is suggested.